# Associations of Dietary Fats with All-Cause Mortality and Cardiovascular Disease Mortality among Patients with Cardiometabolic Disease

**DOI:** 10.3390/nu14173608

**Published:** 2022-08-31

**Authors:** Tingting Yang, Jing Yi, Yangting He, Jia Zhang, Xinying Li, Songqing Ke, Lu Xia, Li Liu

**Affiliations:** School of Public Health, Tongji Medical College, Huazhong University of Science and Technology, Wuhan 430030, China

**Keywords:** dietary fats, cardiometabolic disease, mortality, mediating effect

## Abstract

Previous studies have shown distinct associations between specific dietary fats and mortality. However, evidence on specific dietary fats and mortality among patients with cardiometabolic disease (CMD) remains unclear. The aim of this study was to estimate the association between consumption of specific fatty acids and survival of patients with CMD and examine whether cardiometabolic biomarkers can mediate the above effects. The study included 8537 participants with CMD, from the Third National Health and Nutrition Examination Survey (NHANES III) and NHANES 1999–2014. Cox proportional hazards regression, restricted cubic spline regression, and isocaloric substitution models were used to estimate the associations of dietary fats with all-cause mortality and cardiovascular disease (CVD) mortality among participants with CMD. Mediation analysis was performed to assess the potential mediating roles of cardiometabolic biomarkers. During a median follow-up of 10.3 years (0–27.1 years), 3506 all-cause deaths and 882 CVD deaths occurred. The hazard ratios (HRs) of all-cause mortality among patients with CMD were 0.85 (95% confidence interval (CI), 95% CI, 0.73–0.99; *p* trend = 0.03) for ω-6 polyunsaturated fatty acids (ω-6 PUFA), 0.86 (95% CI, 0.75–1.00; *p* trend = 0.05) for linoleic acid (LA), and 0.86 (95% CI, 0.75–0.98; *p* trend = 0.03) for docosapentaenoic acid (DPA). Isocalorically replacing energy from SFA with PUFA and LA were associated with 8% and 4% lower all-cause mortality respectively. The HRs of CVD mortality among CMD patients comparing extreme tertiles of specific dietary fats were 0.60 (95% CI, 0.48–0.75; *p* trend = 0.002) for eicosapentaenoic acid (EPA), and 0.64 (95% CI, 0.48–0.85; *p* trend = 0.002) for DPA and above effects were mediated by levels of total cholesterol (TC), triglycerides (TG), low density lipoprotein cholesterol (LDL), and high density lipoprotein cholesterol (HDL). Restricted cubic splines showed significant negative nonlinear associations between above specific dietary fats and mortality. These results suggest that intakes of ω-6 PUFA, LA, and DPA or replacing SFA with PUFA or LA might be associated with lower all-cause mortality for patients with CMD. Consumption of EPA and DPA could potentially reduce cardiovascular death for patients with CMD, and their effects might be regulated by cardiometabolic biomarkers indirectly. More precise and representative studies are further needed to validate our findings.

## 1. Introduction

Cardiometabolic disease (CMD), mainly consisting of cardiovascular disease (CVD) and diabetes mellitus [1,2,3], are a leading contributor to death and disability in the world [4,5,6,7]. There were estimated 451 million adults living with diabetes worldwide in 2017 [8]. The prevalence of CVD increased from 271 million in 1990 to 523 million in 2019 [9]. Furthermore, CVD has been the leading cause of death globally. An estimated 17.9 million people die from CVD each year, accounting for 31% of all deaths worldwide [10]. Among various environmental exposures, diet is one of the most important modifiable risk factors for CMD [11], especially dietary fats [12].

Depending on the saturability in the carbon chain, fatty acid can be divided into three classes [13]: (1) saturated fatty acids (SFA), with no double bond; (2) monounsaturated fatty acids (MUFA), with only one double bond; (3) polyunsaturated fatty acids (PUFA), with two or more double bonds. The PUFA can be divided into ω-6 PUFA and ω-3 PUFA. The ω-6 PUFA is mainly composed of linoleic acid (LA), the most abundant ω-6 PUFA, and arachidonic acid (AA). According to food sources of ω-3 PUFA, it can be classified as plant-derived ω-3 PUFA (alpha linolenic acid (ALA)), the most abundant ω-3 PUFA and marine food-derived ω-3 PUFA (marine ω-3 PUFA). The marine ω-3 PUFA also includes docosahexaenoic acid (DHA), eicosapentaenoic acid (EPA), and docosapentaenoic acid (DPA).

Previous studies have shown divergent associations between specific dietary fats and mortality. A recent meta-analysis of prospective cohort studies found that MUFA and PUFA were inversely associated with all-cause mortality and higher intake of PUFA can reduce CVD mortality [14]. Cohort studies also found higher intake of ω-6 PUFA, especially LA and ω-3 PUFA can lower all-cause mortality and higher intake of marine ω-3 PUFA can reduce CVD mortality [12,15]. Furthermore, a cross-sectional analysis from the PURE study concluded that intake of total fat and each type of fat were associated with higher concentrations of total cholesterol (TC), total triglyceride (TG), and low density lipoprotein cholesterol (LDL), but also with higher high density lipoprotein cholesterol (HDL) [16]. It suggests that the health effect of dietary fats might be regulated by potential cardiometabolic biomarkers. However, the association between dietary fats and mortality among CMD patients remains unclear [17,18]. To estimate the survival effects of dietary fats and the mediation role of cardiometabolic biomarkers among patients with prevalent CMD, the current study was conducted in patients with prevalent CMD from the National Health and Nutrition Examination Survey (NHANES) III and NHANES 1999–2014. We hypothesized that intakes of specific dietary fats were associated with all-cause mortality and CVD mortality among patients with CMD, and such effects might be mediated by cardiometabolic biomarkers.

## 2. Materials and Methods

### 2.1. Study Population

Detailed information of NHANES has been described elsewhere [19,20]. Briefly, NHANES is a major program of the National Center for Health Statistics (NCHS) conducted annually for the purpose of assessing the health and nutritional status of people in the United States by interviews and physical examinations. The complex and multi-stage survey examines about 5000 persons each year from various counties across the country, and they were assigned weights for their probability of being randomly selected into the study. To represent the U.S. population, their weighs were taken into account in our analysis. In our current study, we used data available on website from NHANES III and NHANES 1999–2014. Disease information was collected from self-reported questionnaires. CVD was defined if participants reported they had congestive heart failure, coronary heart disease, angina, heart attack, or stroke. Diabetes mellitus was defined if participants had self-reported diabetes or taking medication (including diabetes pills or insulin) to lower blood sugar at baseline. In this analysis, 11,249 patients with CMD were included (4428 with CVD, 4961 with diabetes only, and 1860 with CVD and diabetes simultaneously). Of the 11,124 patients with CMD, participants were excluded because: (a) they had extreme energy intake (total energy intake <800 kcal/day or >4200 kcal/day for men, and <500 kcal/day or >3500 kcal/day for women) (*n* = 395), (b) they had missing information on fatty acids consumption and covariates (*n* = 1252), or (c) they had no mortality follow-up (*n* = 1065). Thus, the remaining analysis consisted of 8537 individuals. The detailed inclusion and exclusion criteria have been presented in Figure 1.

### 2.2. Dietary Assessment and Other Covariates

Our dietary data were collected from dietary recall from NHANES III and NHANES 1999–2014. In NHANES III and NHANES 1999–2002, one piece of 24 h diet data were collected to estimate intakes of food during the 24-h period prior to the interview (midnight to midnight), and the method has been validated as a useful method for dietary assessment [21]. After 2002, the second 24 h diet data were collected by telephone 3 to 10 days later. An average of two pieces of 24 h dietary intake data during NHANES 2003–2014 was used to represent usual dietary intake status of participants, and it is a more accurate estimation to use a minimum of two nonconsecutive days of dietary intake data [22]. Based on obtained dietary consuming information, intakes of total energy and nutrients were calculated based on the United States Department of Agriculture (USDA) Survey Nutrient Database [23,24].

Other covariates were obtained by questionnaires, examinations, and laboratory tests, including gender, age, race (non-Hispanic white, non-Hispanic black, Mexican American, and others), education level (below high school, accomplished high school and higher than high school), family income-poverty ratio (tertiles of PIR, <1.3, 1.3–2.4, ≥2.4), Body Mass Index (BMI, <18.5, 18.5–30, ≥30), use of medication to lower blood sugar, self-reported health status, baseline history of hypertension, hyperlipidemia and cancer, total cholesterol (TC), triglycerides (TG), high density lipoprotein (HDL), and low density lipoprotein (LDL). Never smoking was defined in the questionnaire as smoking fewer than 100 cigarettes in life. Participants who had smoked over 100 cigarettes during their lifetime and reported smoking at the time of the interview were considered as current smokers. Former smokers were defined as those who smoked over 100 cigarettes and had quit smoking. Alcohol drinking status was categorized as none drinking (0 g/day), moderate drinking (0.1 to 27.9 g/day for men and 0.1 to 13.9 g/day for women), and heavy drinking (28 g/day or more for men and 14 g/day or more for women). Physical activity was classified as inactive, insufficiently active, and active according to tertiles of metabolic equivalent of task (MET).

### 2.3. Ascertainment of Death

Mortality status was determined by probabilistic matching to the National Death Index (NDI) through 31 December 2015 based on a unique study identifier [25]. Briefly, the linkage between the survey data and the NDI combined both deterministic and probabilistic approaches. The deterministic approach is based on primary identifiers of Social Security number 9 (SSN9) or Social Security number 4 (SSN4) (depending on the survey year or cycle of the survey), first name, middle initial, last name, or father’s surname, race, sex, and so on. The probabilistic approach performs weighting and link adjudication following the Fellegi–Sunter method, which is the foundational methodology used for record linkage, showing a high accuracy by a prior validation study [26]. The underlying causes-of-death were classified based on the international statistical classification of diseases, 10th revision (ICD-10). In addition, data on mortality caused by stroke (I60–I69) were available only until 31 December 2011. Therefore, the main outcomes of our study were all-cause mortality and CVD (not including stroke) mortality.

### 2.4. Statistical Analyses

All analyses accounted for the complex survey design, including sample weights, clustering and stratification [19]. In our analyses, the amount of macronutrients intake was represented as percent of total energy (%E) by the nutrient density method, which was computed by dividing energy from the nutrient by the total daily energy intake (%E = ((fats (g) × 4)/total energy intake (kcal)) × 100) [27]. We categorized participants into tertiles of intake levels. Cox proportional hazards regression was applied to investigate the associations of dietary fats with all-cause and CVD mortality among patients with CMD. Person-years of follow-up were calculated from baseline to the earliest of time of death, loss to or unavailability for follow-up, or the end of follow-up, whichever came first. Hazard ratios (HRs) and the corresponding 95% confidence intervals (CIs) were calculated by using the lowest tertile group as reference. Tests for trends were performed by assigning median values to corresponding categories of intakes as continuous variables, which were adjusted for all potential confounding variables. Additionally, restricted cubic splines with three knots were used to detect dose–response relationships between dietary fats and mortality. Furthermore, we evaluated the effect of replacing SFA with other types of fats by establishing isocaloric substitution models that simultaneously included total energy intake and percentages of energy from proteins, carbohydrates, and all fats except SFAs. By leaving SFAs out of the model, regression coefficients of the other fats could be interpreted as estimated effects of replacing SFAs with specific fats while leaving other fatty acids unchanged. Mediation analysis was performed to assess the mediating roles of cardiometabolic biomarkers (including total cholesterol (TC), total triglyceride (TG), low density lipoprotein cholesterol (LDL), and high density lipoprotein cholesterol (HDL)) and beneficial fatty acids in CVD mortality among participants with CMD. Furthermore, we also conducted additional analysis to explore the association of specific dietary fats and all-cause mortality and CVD mortality in the general population.

Sensitivity analyses were conducted to test the robustness of our results, including evaluating the associations of dietary fats with all-cause and CVD mortality among CMD patients after excluding patients who died within the first two years of follow-up and further adjustment of the use of medication (including diabetes pills and insulin) from NHANES III and NHANES 1999–2014. All statistical analyses were conducted using SAS software version 9.4 (SAS Institute, Cary, NC, USA) and R software. Mediation analysis was generated by SAS Mediation Macro. A two-sided *p*-value < 0.05 was considered statistically significant.

## 3. Results

### 3.1. Population Characteristics

During a median follow-up length of 10.3 years (0–27.1 years), 3506 (41.07%) deaths occurred, including 882 deaths from CVD. Table 1 shows baseline characteristics of patients with CMD in NHANES III and NHANES 1999–2014. Among 8537 patients with CMD, participants with higher SFA, MUFA, and PUFA intakes were more likely to be non-Hispanic white, richer, and fatter. However, they were less likely to drink. The participants with higher consumption of SFA and MUFA were more likely to be male, younger, current smokers, and with good self-reported health status. For participants consuming more PUFA, they tended to be higher educated, and with more prevalent hypertension and hyperlipidemia. Intakes of SFA, MUFA, and PUFA were positively associated with total energy intake but inversely related to carbohydrate intake. In addition, those with higher PUFA intake seemed to have more protein consumption (Table 1). For specific PUFA, the baseline characteristics of patients with CMD according to the tertiles of energy percentage from ω-6 PUFA and ω-3 PUFA are presented in the Appendix A. The participants with more ω-6 PUFA consumption had the similar baseline characteristics with those having more consumption of PUFA. Those with higher ω-3 PUFA consumption were more likely to be higher educated, richer, and with good self-reported health status. They also tended to consume more total energy but less carbohydrates (Appendix A).

### 3.2. All-Cause Mortality

Among patients with CMD, total SFA, MUFA, and PUFA were not associated with all-cause mortality, but ω-6 PUFA, and the majority ω-6 PUFA, LA were inversely associated with all-cause mortality in multivariable-adjusted models, with the HRs of 0.85 (95% CI: 0.73–0.99; *p* trend = 0.03) and 0.86 (95% CI: 0.75–1.00; *p* trend = 0.03) for highest vs. lowest tertile, respectively. Similarly, DPA were inversely associated with all-cause mortality with the HRs of 0.86 (95% CI: 0.75–0.98; *p* trend = 0.03) for highest vs. lowest tertile, respectively (Table 2). No association of residual fatty acids components with all-cause mortality was observed (Table 2). In the substitution model, a 5% replacement of energy from SFA with PUFA, and a 2% replacement of energy from SFA with LA were associated with an 8% lower all-cause mortality and a 4% lower all-cause mortality, respectively (Figure 2). Figure 3 showed a nonlinear association between ω-6 PUFA, LA, DPA, and all-cause mortality (all *p* nonlinear < 0.05). The HRs for all-cause mortality decreased as the intake of above dietary fats increased and reached a plateau when intakes were approximately 7.7%E, 7.5%E, and 0.02%E for ω-6 PUFA, LA, and DPA, respectively. This suggests that once dietary fats reach a certain level, the all-cause mortality hardly decreases with further increases in ω-6 PUFA, LA, and DPA. For the general population, the intakes of ω-6 PUFA, ω-3 PUFA, marine ω-3 PUFA, and its components (DHA, EPA, DPA) were inversely associated with all-cause mortality. (Appendix A).

### 3.3. CVD Mortality

Among patients with CMD, total SFA, MUFA, and PUFA were not associated with CVD mortality. For specific PUFAs, marine ω-3 PUFA components-EPA and DPA were strongly associated with reduced CVD mortality, with HRs for highest vs. lowest tertile of 0.60 (95% CI, 0.48–0.75; *p* trend = 0.002) and 0.64 (95% CI, 0.48–0.85; *p* trend = 0.002), respectively (Table 3). No association of residual PUFA components with CVD mortality was observed (Table 3). The nonlinear dose–response relationships of EPA and DPA with CVD mortality among patients with CMD are shown in Figure 3. A U-shaped curve was observed between EPA and CVD mortality. The CVD mortality decreased dramatically along with the increased intakes of EPA at a low level and then kept rising at a relatively higher level. The protective effect for CVD mortality approached to saturation when EPA accounted for 0.08%E. For DPA in CVD mortality, we can see the same trend as in all-cause mortality. Mediating effect analysis indicated that the health effect of EPA and DPA in CVD mortality could be regulated by cardiometabolic biomarkers through indirect effects with different mediating proportions. When higher intake of EPA was compared with lower intake of EPA, the proportion mediated by cardiometabolic biomarkers was 5.33% for TC, and 6.18% for TG, 1.84% for HDL, and 3.94% for LDL (Table 4). Similarly, when higher intake of DPA was compared with lower intake of DPA, the proportion mediated by cardiometabolic biomarkers was 9.55% for TC, and 6.53% for TG, 2.74% for HDL, and 11.12% for LDL (Table 4). For the general population, the intakes of ω-3 PUFA, marine ω-3 PUFA, and its components (DHA, EPA, DPA) were inversely associated with CVD mortality (Appendix A).

### 3.4. Sensitivity Analyses

We observed similar relationships of dietary fatty acids with CVD mortality among two sensitivity analyses (Appendix A). For all-cause mortality, we can only see the protective effect of DPA, not including ω-6 PUFA and LA after excluding patients who died within the first two years of follow-up (Appendix A). After further adjustment of use of medication (including diabetes pills or insulin), the protective effect of DPA disappeared but similar associations of ω-6 PUFA and LA with all-cause mortality still maintained (Appendix A).

## 4. Discussion

In this study, we investigated the associations between intakes of dietary fatty acids with all-cause mortality and CVD mortality among representative US patients with CMD from NHANES III and NHANES 1994–2014. The ω-6 PUFA, ω-6 PUFA-LA, and marine ω-3 PUFA-DPA were associated with reduced all-cause mortality risk. The components of marine ω-3 PUFA (EPA and DPA) were strongly associated with a reduced risk of CVD mortality among patients with CMD. The findings underscore the importance of maintaining PUFA consumption, especially marine ω-3 PUFA, to reduce the risk of CVD mortality in CMD patients.

To our knowledge, this is the first prospective analysis that investigated the associations between specific dietary fats and all-cause mortality and CVD mortality among people with CMD. In our study, ω-6 PUFA (accounting for 88.72% of total PUFA) and LA (accounting for 98.86% of ω-6 PUFA) were related with reduced all-cause mortality among CMD patients. Substitution analyses also revealed the protective effect of PUFA and LA in all-cause mortality. The dose–response relationship clearly sees the declining trend with increasing intake of ω-6 PUFA and LA. Although similar research about effects of fatty acids and mortality is scarce in the CMD population, studies on CVD or diabetes mellitus could provide related information. A review from randomized controlled trials summarized that the replacement of SFA with ω-6 PUFA had potential benefits on CVD incidence and mortality [28]. A previous study by Nagai et al. indicated that lower circulating ω-6 PUFA levels were significantly associated with all-cause death in patients with acute decompensated heart failure [29]. A meta-analysis by Wu et al. concluded that LA had long-term benefits for the prevention of type 2 diabetes (T2D) [30]. Furthermore, two prospective, longitudinal cohort studies also found inverse associations of the intake of LA with all-cause mortality among patients with T2D [17]. However, a randomized controlled trial concluded that LA led to increased mortality among patients with coronary heart disease [31]. The observation might be confounded by high trans-fat contents of margarines, which were used as the food sources of LA in this intervention [32]. It reminds us that more accurate and effective intervention studies are needed to validate our findings. It is worth noting that, when we excluded patients who died within the first two years of follow-up to reduce selecting bias, we found that the protective effects of ω-6 PUFA and LA disappeared. It warns us to draw conclusions about the above associations cautiously.

In the current study, we found that dietary DPA could lead to decreased all-cause mortality among CMD patients. However, after further adjustment of the use of medication (including diabetes pills or insulin), the protective effect of DPA disappeared. We also observed reduced CVD mortality among CMD patients with higher intake of EPA and DPA (components of marine ω-3 PUFA). Numerous studies have recognized the potential cardiovascular benefits of marine ω-3 PUFA, including DHA, EPA, and DPA [33,34]. Our findings are consistent with a meta-analysis based on 13 random clinical trials which concluded that marine ω-3 fatty acids were associated with reduced CVD death, with the HR of 0.92 (95% CI: 0.88–0.97) compared with placebo [35]. Moreover, a multicenter prospective cohort also revealed that the circulating levels of ω-3 PUFA (including DHA, EPA, and DPA) were inversely associated with all-cause mortality, especially the CHD death in older adults [36]. Two cohort studies in the United States concluded that marine ω-3 PUFA could reduce CVD mortality among patients with type 2 diabetes [17]. A randomized controlled trial also found in patients with a history of coronary artery disease who were given EPA treatment, and major coronary events were reduced by 19% in the EPA supplement group vs. in the control group [37]. A total of 67 prospective studies that is comprised of 310,955 participants identified that EPA and DPA could lead to a lower risk of T2D [38]. For EPA, the mechanisms of the cardiac-protect effects may include decreasing plasma triglyceride, resting heart rate and blood pressure, inhibiting platelet activity, and stabilizing/regressing of coronary plaque [35,39,40,41,42,43]. Compared with EPA, research concerning the protective effects of DPA in CVD development is limited. A study combining 19 cohort studies reported that DPA could reduce the risk of coronary heart disease in the general population [44]. The potential mechanisms involved include anti-inflammation, inhibition of cytokine synthesis, anti-thrombosis, and improvement of plasma lipids [36,45,46,47,48]. However, we did not observe a significant association between ALA and CVD mortality among patients with CMD, which accounted for 92.48% of total ω-3 PUFA. Epidemiological studies concerning cardiovascular benefits of ALA remain mixed [49,50,51], indicating that more precise and representative studies are needed. In mediating analysis, the protective effect of EPA and DPA in CVD mortality can work through TC, TG, HDL, and LDL indirectly. Many studies have identified that ω-3 PUFA could influence the concentrations of TC, TG, LDL, and HDL [52,53,54]. However, how the potential influence of these bioactive lipids is regulated by marine ω-3 PUFA is still unclear.

Our study has several strengths. First, the prospective study design, well-controlled and rigorous protocol, large sample size, long follow-up period, and relatively accurate estimates ensured the validity of the current study. Second, besides categorizing intakes of dietary fatty acids into tertiles, we also regarded the dietary fatty acids as continuous variables to estimate the dose–response relationships of fatty acids with mortality. Third, besides ω-6 PUFA and ω-3 PUFA, we further estimated the protective effects of their specific components. However, there are still some limitations. First, dietary data were obtained by recalling, so inaccuracies may exist. However, well trained workers, an abundant dietary database from USDA, and the reliable method and system ensured that the measures of diets were as accurate as possible. Second, dietary behaviors may be altered during follow up. Although day-to-day variations could not be estimated, we have used the average intake level to represent their usual intake status to minimize the measurement errors. Third, even though we adjusted many confounders in regression models, residual confounding effects from unmeasured or unknown factors could not be excluded entirely such as the quality of diabetes management or insulin resistance. More precise and representative studies are further needed. Fourth, mortality status was based on the deterministic and probabilistic matching to the NDI, which might result in misclassification. However, the linkage algorithm can produce high-quality matches with a low degree of linkage error [26]. Lastly, the study was an observational study, and we could not verify causality.

## 5. Conclusion

These results suggest that intakes of ω-6 PUFA, LA, and DPA or replacing SFA with PUFA or LA might be associated with lower all-cause mortality for patients with CMD. Consumption of EPA and DPA could potentially reduce cardiovascular death for patients with CMD, and their effects might be regulated by cardiometabolic biomarkers indirectly. More precise and representative studies are further needed to validate our findings.

## Figures and Tables

**Figure 1 nutrients-14-03608-f001:**
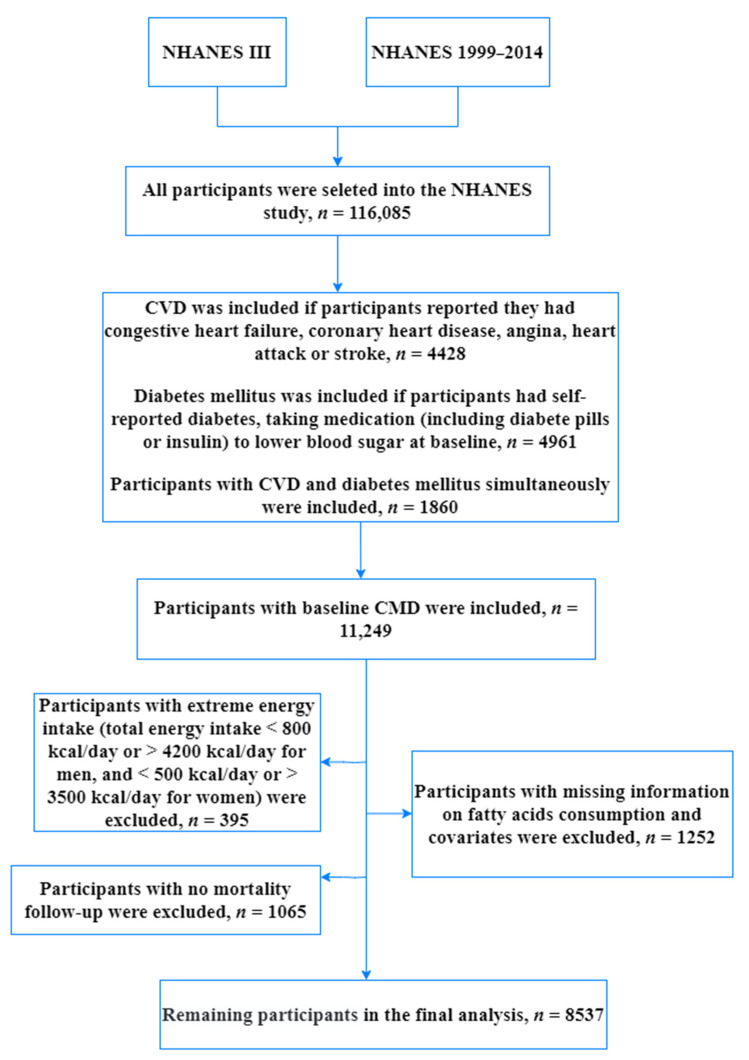
The flow chart of selecting a full analysis set.

**Figure 2 nutrients-14-03608-f002:**
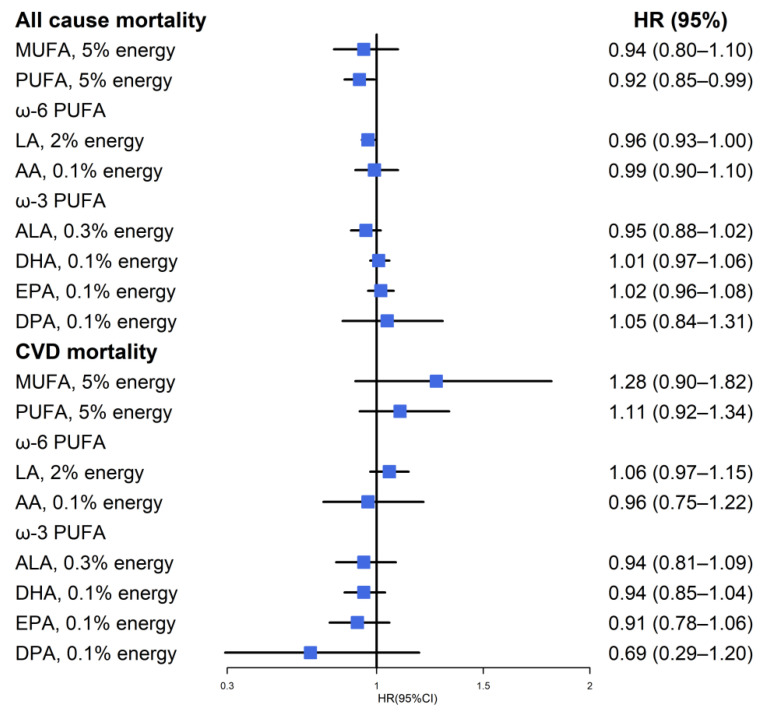
Multivariable hazard ratios (HRs) of all cause mortality and CVD mortality by isocaloric substitution of specific fatty acid for saturated fatty acids among participants with CMD in NHANESIII and NHANES 1999–2014.

**Figure 3 nutrients-14-03608-f003:**
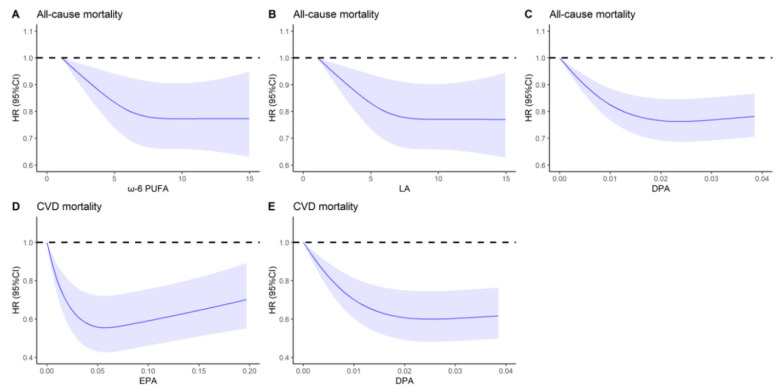
Dose–response relationships of effective dietary acids with all-cause and CVD mortality among patients with CMD in NHANES III and NHANES 1999–2014 ^a^. ^a^: Dose–response relationships between effective dietary acids with all-cause and CVD mortality among patients with CMD, (**A**) ω-6 PUFA in all-cause mortality (*p* nonlinear = 0.0454), (**B**) LA in all-cause mortality (*p* nonlinear = 0.0447), (**C**) DPA in all-cause mortality (*p* nonlinear < 0.0001), (**D**) EPA in CVD mortality (*p* nonlinear < 0.0001), (**E**) DPA in CVD mortality (*p* nonlinear < 0.0001) in NHANES III and NHANES 1999–2014 were estimated by multivariable adjusted Cox regression models based on restricted cubic splines. Solid splines represented estimated hazard ratios, and cloud areas represented the corresponding 95% CI. The multivariable-adjusted model was adjusted for gender, age, race, educational levels, PIR, BMI, smoking status, alcohol drinking status, physical activity status, self-reported health status, baseline history of hypertension, hyperlipidemia and cancer, total energy intake, energy intake derived from carbohydrate, protein intake (in tertiles), cholesterol levels (in tertiles), and energy intake derived from remaining fatty acids (SFA, MUFA, and PUFA) where appropriate. Abbreviation and acronyms: CMD, cardiometabolic disease; CVD, cardiovascular disease; ω-6 PUFA, ω-6 polyunsaturated fatty acids; LA, linoleic acid; EPA, eicosapentaenoic acid; DPA, docosapentaenoic acid.

**Table 1 nutrients-14-03608-t001:** Baseline characteristics of patients with cardiometabolic diseases in NHANES III and NHANES 1999–2014 according to the tertiles of energy percentage from SFA, MUFA, and PUFA ^a^.

Characteristics	Tertile of SFA Intake			Tertile of MUFA Intake			Tertile of PUFA Intake		
	T1	T2	T3	T1	T2	T3	T1	T2	T3
Participants (*n*)	2840	2851	2846	2845	2846	2846	2845	2849	2843
Fatty acid intake (% of energy)	7.19	10.77	14.92	8.58	12.29	16.65	4.47	7.03	10.67
Male (%)	45.05	49.17	50.90	42.57	47.53	54.40	46.97	48.68	49.92
Age (years)	62.38	61.76	59.98	62.37	62.32	59.57	61.33	61.92	60.74
Race (%)									
Non-Hispanic White	63.76	72.69	76.75	67.45	69.70	76.18	70.40	70.79	72.96
Non-Hispanic Black	15.10	13.57	12.17	12.94	14.07	13.59	12.50	14.53	13.64
Mexican American	7.47	6.07	5.07	7.03	6.80	4.83	6.25	6.63	5.57
Others	13.68	7.67	6.014	12.58	9.43	5.40	10.85	8.05	7.83
Education level (%)									
Less than high school	35.41	31.66	31.63	35.73	31.32	31.57	37.07	33.97	27.53
High school or equivalent	26.76	28.87	27.64	25.74	28.36	29.00	26.07	29.30	28.05
College or above	37.83	39.47	40.73	38.53	40.32	39.43	36.87	36.73	44.42
Family income-poverty ratio level (%)									
<1.3	30.39	22.58	23.58	31.00	25.81	20.16	29.05	25.13	21.87
1.3–2.4	30.44	33.37	30.89	31.03	32.11	31.59	31.36	33.72	29.82
≥2.4	39.17	44.05	45.53	37.96	42.08	48.25	39.59	41.15	48.31
BMI group (%)									
<18.5	1.66	1.12	0.67	1.456	1.46	1.05	0.86	1.934	0.64
18.5–25	23.27	18.90	17.97	22.73	22.73	17.43	21.89	20.30	17.59
25–30	31.73	31.50	30.95	31.66	31.66	30.43	33.06	29.64	31.31
≥30	43.34	48.48	50.41	44.15	44.15	51.10	44.19	48.13	50.46
Smoking status (%)									
Never smoker	45.50	42.48	39.61	46.10	43.91	37.93	42.11	42.54	42.50
Former smoker	38.10	37.84	37.75	36.53	36.61	40.10	37.20	37.06	39.31
Current smoker	16.40	19.68	22.64	17.37	19.48	21.97	20.69	20.40	18.19
Alcohol drinking (%)									
Non-drinker	80.17	79.48	82.00	78.58	80.74	82.18	79.78	78.94	82.93
Low to moderate drinker	10.95	12.69	12.86	11.59	12.45	12.55	11.18	14.58	11.06
Heavy drinker	8.88	7.83	5.14	9.83	6.81	5.27	9.04	6.48	6.01
Physical activity (%)									
Inactive	39.78	39.41	42.20	40.12	40.95	40.51	38.11	39.92	43.47
Insufficiently active	33.05	33.41	34.27	33.02	32.32	35.19	34.98	34.19	31.72
Active	27.18	27.19	23.53	26.86	26.73	24.30	26.91	25.89	24.81
Self-reported health status (%)									
Poor to fair	44.29	43.29	42.37	45.99	43.08	41.12	46.88	41.22	41.58
Good	35.79	33.62	38.49	31.30	37.21	39.06	32.23	37.91	38.08
Very good/excellent	19.91	23.09	19.14	22.71	19.71	19.82	20.89	20.87	20.34
Self-reported chronic diseases (%)									
Hypertension	61.53	66.05	39.03	61.18	64.13	63.13	59.66	64.65	64.29
Hyperlipemia	69.14	70.11	67.37	68.70	71.07	67.04	64.75	69.98	71.80
Cancer	17.70	19.02	17.34	17.41	19.49	17.27	16.26	19.98	17.92
Total energy intake (kcal/day)	1624.05	1840.65	1938.68	1601.17	1820.33	1975.98	1694.55	1798.49	1932.44
Protein intake (% of energy)	16.79	16.42	16.75	16.90	16.51	16.56	17.20	16.81	15.97
Carbohydrate intake (% of energy)	56.61	49.02	42.50	57.53	49.50	41.40	53.43	49.03	44.64

^a^ All estimates accounted for complex survey designs. Values were mean for continuous variables and percentages for categorical variables. Abbreviation and acronyms: BMI body mass index.

**Table 2 nutrients-14-03608-t002:** Associations of specific fatty acids with all-cause mortality among patients with CMD ^a^.

	Tertiles of Percentage Energy from Specific Fatty Acids			
	Tertile 1	Tertile 2	Tertile 3	*p* Trend ^b^
SFA				
Mean, % of energy	7.19 ± 0.05	10.77 ± 0.03	14.92 ± 0.08	
No. of deaths/person years	1180/287,779	1147/269,766	1179/287,119	
Model 1 ^c^	1 (ref.)	0.92 (0.81–1.06)	0.99 (0.90–1.09)	0.83
Model 2 ^d^	1 (ref.)	0.94 (0.81–1.09)	0.99 (0.84–1.15)	0.94
MUFA				
Mean, % of energy	8.58 ± 0.05	12.29 ± 0.02	16.65 ± 0.11	
No. of deaths/person years	1208/279,237	1099/260,626	1199/304,801	
Model 1 ^c^	1 (ref.)	0.99 (0.88–1.13)	0.99 (0.88–1.12)	0.93
Model 2 ^d^	1 (ref.)	1.09 (0.93–1.26)	1.08 (0.86–1.35)	0.56
PUFA				
Mean, % of energy	4.49 ± 0.03	7.03 ± 0.02	10.67 ± 0.07	
No. of deaths/person years	1406/306,092	1106/269,233	984/269,339	
Model 1 ^c^	1 (ref.)	0.96 (0.84–1.08)	0.90 (0.80–1.03)	0.07
Model 2 ^d^	1 (ref.)	0.94 (0.82–1.06)	0.88 (0.76–1.02)	0.06
ω-6 PUFA				
Mean, % of energy	3.92 ± 0.03	6.26 ± 0.02	9.58 ± 0.07	
No. of deaths/person years	1413/302,958	1091/270,768	1002/270,938	
Model 1 ^c^	1 (ref.)	0.90 (0.79–1.03)	0.86 (0.74–0.99)	0.07
Model 2 ^d^	1 (ref.)	0.86 (0.74–1.01)	0.85 (0.73–0.99)	0.03
LA				
Mean, % of energy	3.86 ± 0.03	6.17 ± 0.02	9.49 ± 0.07	
No. of deaths/person years	1409/303,023	1101/270,272	996/271,369	
Model 1 ^c^	1 (ref.)	0.92 (0.80–1.05)	0.89 (0.78–1.01)	0.06
Model 2 ^d^	1 (ref.)	0.92 (0.80–1.05)	0.86 (0.75–1.00)	0.05
AA				
Mean, % of energy	0.02 ± 0.0004	0.06 ± 0.0003	0.13 ± 0.0001	
No. of deaths/person years	1250/287,909	1078/274,846	1178/281,909	
Model 1 ^c^	1 (ref.)	0.94 (0.83–1.06)	0.98 (0.86–1.12)	0.80
Model 2 ^d^	1 (ref.)	0.96 (0.85–1.09)	1.05 (0.89–1.23)	0.51
ω-3 PUFA				
Mean, % of energy	0.41 ± 0.003	0.65 ± 0.001	1.12 ± 0.012	
No. of deaths/person years	1430/321,955	1145/271,423	931/251,286	
Model 1 ^c^	1(ref.)	1.04(0.93–1.15)	0.91(0.79–1.03)	0.12
Model 2 ^d^	1(ref.)	1.04(0.93–1.17)	0.90(0.89–1.04)	0.15
ALA				
Mean, % of energy	0.37 ± 0.003	0.59 ± 0.001	1.00 ± 0.010	
No. of deaths/person years	1422/320,237	1143/276,767	941/247,660	
Model 1 ^c^	1 (ref.)	1.04 (0.92–1.18)	0.94 (0.81–1.09)	0.36
Model 2 ^d^	1 (ref.)	1.02 (0.89–1.17)	0.93 (0.80–1.09)	0.36
Marine ω-3 PUFA				
Mean, % of energy	0.002 ± 0.0001	0.022 ± 0.0003	0.19 ± 0.008	
No. of deaths/person years	1638/342,543	829/227,727	1039/274,394	
Model 1 ^c^	1 (ref.)	0.92 (0.80–1.06)	0.92 (0.80–1.06)	0.29
Model 2 ^d^	1 (ref.)	0.91 (0.78–1.07)	0.97 (0.84–1.12)	0.83
DHA				
Mean, % of energy	0.0005 ± 0.00003	0.01 ± 0.0001	0.11 ± 0.004	
No. of deaths/person years	1549/326,118	872/233,996	1053/284,550	
Model 1 ^c^	1 (ref.)	0.92 (0.80–1.05)	0.93 (0.81–1.07)	0.34
Model 2 ^d^	1 (ref.)	0.92 (0.79–1.06)	0.99 (0.86–1.14)	0.98
EPA				
Mean, % of energy	0.00003 ± 0.000003	0.003 ± 0.00005	0.06 ± 0.003	
No. of deaths/person years	1848/385,458	771/208,040	887/251,166	
Model 1 ^c^	1 (ref.)	0.99 (0.84–1.16)	0.92 (0.80–1.06)	0.19
Model 2 ^d^	1 (ref.)	0.95 (0.79–1.13)	0.93 (0.80–1.07)	0.36
DPA				
Mean, % of energy	0 ± 0	0.004 ± 0.00007	0.03 ± 0.0010	
No. of deaths/person years	2024/418,989	714/195,684	768/229,991	
Model 1 ^c^	1 (ref.)	0.84 (0.72–0.96)	0.85 (0.74–0.98)	0.02
Model 2 ^d^	1 (ref.)	0.80 (0.69–0.93)	0.86 (0.75–0.98)	0.03

^a^ Values were presented as mean ± standard error for energy percentage from specific fatty acids. All estimates accounted for complex survey designs. ^b^
*p* trend was calculated by median values of each fatty acids. ^c^ Model 1 was adjusted for gender, age, race, educational levels, and PIR. ^d^ Model 2 was adjusted for gender, age, race, educational levels, PIR, BMI, smoking status, alcohol drinking status, physical activity status, self-reported health status, baseline history of hypertension, hyperlipidemia, total energy intake, energy intake derived from carbohydrate, protein intake (in tertiles), cholesterol levels (in tertiles), and energy intake derived from remaining fatty acids (SFA, MUFA, and PUFA) where appropriate.

**Table 3 nutrients-14-03608-t003:** Associations of specific fatty acids with CVD mortality among patients with CMD ^a^.

	Tertiles of Percentage Energy from Specific Fatty Acids			
	Tertile 1	Tertile 2	Tertile 3	*p*-Trend ^b^
SFA				
Mean, % of energy	7.19 ± 0.05	10.77 ± 0.03	14.92 ± 0.08	
No. of deaths/person years	292/287,779	290/269,766	300/287,119	
Model 1 ^c^	1 (ref.)	0.76 (0.55–1.05)	0.85 (0.64–1.11)	0.25
Model 2 ^d^	1 (ref.)	0.84 (0.60–1.19)	0.93 (0.65–1.34)	0.65
MUFA				
Mean, % of energy	8.58 ± 0.05	12.29 ± 0.02	16.6 5 ± 0.11	
No. of deaths/person years	316/279,237	260/260,626	306/304,801	
Model 1 ^c^	1 (ref.)	0.86 (0.69–1.09)	0.94 (0.75–1.18)	0.50
Model 2 ^d^	1 (ref.)	0.99 (0.78–1.27)	1.18 (0.85–1.65)	0.27
PUFA				
Mean, % of energy	4.49 ± 0.03	7.03 ± 0.02	10.67 ± 0.07	
No. of deaths/person years	363/306,092	287/269,233	232/269,339	
Model 1 ^c^	1 (ref.)	1.20 (0.94–1.54)	0.91 (0.70–1.19)	0.32
Model 2 ^d^	1 (ref.)	1.20 (0.95–1.52)	0.92 (0.67–1.25)	0.51
ω-6 PUFA				
Mean, % of energy	3.92 ± 0.03	6.26 ± 0.02	9.58 ± 0.07	
No. of deaths/person years	361/302,958	286/270,768	235/270,938	
Model 1 ^c^	1 (ref.)	1.12 (0.86–1.46)	0.91 (0.69–1.21)	0.23
Model 2 ^d^	1 (ref.)	1.13 (0.87–1.47)	0.91 (0.65–1.26)	0.48
LA				
Mean, % of energy	3.86 ± 0.03	6.17 ± 0.02	9.49 ± 0.07	
No. of deaths/person years	363/303,023	286/270,272	233/271,369	
Model 1 ^c^	1 (ref.)	1.07 (0.83–1.37)	0.92 (0.69–1.24)	0.58
Model 2 ^d^	1 (ref.)	1.09 (0.85–1.39)	0.93 (0.67–1.30)	0.66
AA				
Mean, % of energy	0.02 ± 0.0003	0.06 ± 0.0002	0.13 ± 0.001	
No. of deaths/person years	337/250,636	262/237,907	283/244,070	
Model 1 ^c^	1 (ref.)	0.78 (0.60–1.02)	0.82 (0.67–1.01)	0.07
Model 2 ^d^	1 (ref.)	0.86 (0.64–1.14)	1.02 (0.75–1.37)	0.90
ω-3 PUFA				
Mean, % of energy	0.41 ± 0.003	0.65 ± 0.001	1.12 ± 0.012	
No. of deaths/person years	387/321,955	279/271,423	216/251,286	
Model 1 ^c^	1 (ref.)	1.04 (0.93–1.15)	0.91 (0.79–1.03)	0.12
Model 2 ^d^	1 (ref.)	0.93 (0.74–1.17)	0.78 (0.58–1.03)	0.09
ALA				
Mean, % of energy	0.37 ± 0.003	0.59 ± 0.001	1.00 ± 0.010	
No. of deaths/person years	380/320,237	280/276,767	222/247,660	
Model 1 ^c^	1 (ref.)	0.99 (0.81–1.21)	0.89 (0.68–1.15)	0.36
Model 2 ^d^	1 (ref.)	0.99 (0.80–1.22)	0.90 (0.67–1.22)	0.49
Marine ω-3 PUFA				
Mean, % of energy	0.002 ± 0.0001	0.022 ± 0.0003	0.19 ± 0.008	
No. of deaths/person years	453/342,543	183/227,727	246/274,394	
Model 1	1 (ref.)	0.72 (0.53–0.98)	0.73(0.60–0.90)	0.01
Model 2	1 (ref.)	0.73 (0.53–1.00)	0.80(0.62–1.02)	0.13
DHA				
Mean, % of energy	0.0005 ± 0.00003	0.01 ± 0.0001	0.11 ± 0.004	
No. of deaths/person years	432/326,118	193/233,996	257/284,550	
Model 1 ^c^	1 (ref.)	0.64 (0.49–0.85)	0.75 (0.60–0.94)	0.04
Model 2 ^d^	1 (ref.)	0.65 (0.48–0.88)	0.83 (0.64–1.08)	0.36
EPA				
Mean, % of energy	0.00003 ± 0.000003	0.003 ± 0.00005	0.06 ± 0.003	
No. of deaths/person years	531/385,458	167/208,040	184/251,166	
Model 1 ^c^	1 (ref.)	0.61 (0.48–0.79)	0.60 (0.49–0.74)	<0.0001
Model 2 ^d^	1 (ref.)	0.56 (0.45–0.75)	0.60 (0.48–0.75)	0.002
DPA				
Mean, % of energy	0 ± 0	0.004 ± 0.00007	0.03 ± 0.0010	
No. of deaths/person years	550/418,989	156/195,684	176/229,991	
Model 1 ^c^	1(ref.)	0.66 (0.50–0.86)	0.64 (0.50–0.82)	0.001
Model 2 ^d^	1(ref.)	0.61 (0.46–0.82)	0.64 (0.48–0.85)	0.002

^a^ Values were presented as mean ± standard error for energy percentage from specific fatty acids. All estimates accounted for complex survey designs. ^b^
*p* trend was calculated by median values of each fatty acids. ^c^ Model 1 was adjusted for gender, age, race, educational levels, and PIR. ^d^ Model 2 was adjusted for gender, age, race, educational levels, PIR, BMI, smoking status, alcohol drinking status, physical activity status, self-reported health status, baseline history of hypertension, hyperlipidemia, total energy intake, energy intake derived from carbohydrate, protein intake (in tertiles), cholesterol levels (in tertiles), and energy intake derived from remaining fatty acids (SFA, MUFA, and PUFA) where appropriate.

**Table 4 nutrients-14-03608-t004:** Analysis of total effect, direct effect, indirect effect, and mediating proportion of cardiometabolic biomarkers between beneficial fatty acids and CVD mortality among participants with CMD in NHANES III and NHANES 1999–2014 ^a^.

	Biomarkers	Total Effect	Direct Effect	Indirect Effect	MediatingProportion (%)
EPA	TC	0.62 (0.54–0.73)	0.64 (0.55–0.75)	0.97 (0.94–1.00)	5.33%
	TG	0.62 (0.51–0.76)	0.65 (0.53–0.79)	0.96 (0.93–1.00)	6.18%
	HDL	0.63 (0.54–0.73)	0.63 (0.54–0.74)	0.99 (0.98–1.00)	1.84%
	LDL	0.64 (0.50–0.82)	0.65 (0.51–0.84))	0.98 (0.93–1.03)	3.94%
DPA	TC	0.69 (0.59–0.81)	0.72 (0.61–0.85)	0.96 (0.93–0.99)	9.55%
	TG	0.63 (0.51–0.78)	0.66 (0.53–0.81)	0.96 (0.93–1.00)	6.53%
	HDL	0.70 (0.59–0.82)	0.71 (0.60–0.83)	0.99 (0.97–1.00)	2.74%
	LDL	0.71 (0.54–0.91)	0.74 (0.57–0.96)	0.96 (0.91–1.01)	11.12%

^a^ Values were presented as HR (95% CI) of total effect, direct effect, indirect effect, and the mediating proportion of cardiometabolic biomarkers between beneficial fatty acids and CVD mortality using a mediating analysis which was adjusted for the same potential confounders as the Cox proportional hazards regression.

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
