# Peer review of "Associations of Dietary Fats with All-Cause Mortality and Cardiovascular Disease Mortality among Patients with Cardiometabolic Disease"

_nutrients, 2022, doi:10.3390/nu14173608_

Round 1

Reviewer 1 Report

In the manuscript, the authors attempted to demonstrate the associations of dietary fats with mortality among patients with cardiometabolic disease.  The article in its current form does not meet the requirements of the journal. The present form article has serious flaws in designing and conducting analyses. Nowhere is it clearly stated that the authors did not collect the data themselves, they only analyzed available data collected by someone else for another purpose. No source (specific database) from where the data was obtained. It is unacceptable to draw conclusions about dietary fats and the risk of death based on a one or two-day diet. this is in no way representative. A large number of participants does not change the lack of representativeness. If ascertainment of death were determined by probabilistic matching so is this mean that the authors are not 100% sure that the specific participant in question actually died? No control group. How is it known that the observed results apply only to cardiometabolic disease and not to the general population?

Reviewer 2 Report

The manuscript by Yang et al reported the analysis of the prospective associations between intakes of specific fatty acids and in particular, saturated fatty acids, monounsaturated fatty acids, polyunsaturated fatty acids of n-6 and n-3 series, and all-cause and cardiovascular disease mortality in subjects with cardiometabolic diseases. They found a reduced risk for all-cause mortality and cardiovascular disease mortality for unsaturated fatty acids, and in particular decreased all-cause mortality for the n-6 linoleic acid and the n-3 docosapentaenoic acid, and decreased cardiovascular disease mortality for the n-3 PUFA components-eicosapentaenoic acid and docosapentaenoic acid. The results are interesting for the further understanding of the complex and often controversial role of dietary fatty acids on mortality risk. The manuscript is well written, and results clearly presented. 

I suggest the following minor concerns:

With reference to the baseline characteristic of participants according to the intake of specific fatty acids (Table 1), is there any differences regarding specific PUFAs, i.e. n-6 and n-3 PUFAs?

Can the intake of specific foods or food groups contributing to the specific fatty acids in the population sample analyzed be reported?

The discussion section might benefit for a more deep elaboration of the literature and mostly the possible explanation of results.

More attention to the manuscript formatting should be paid: only as an example, see lines 71-73.

Reviewer 3 Report

The manuscript submitted to Nutrients by Yang et al., titled: "Associations of Dietary Fats with All-cause Mortality and Cardiovascular Disease Mortality among Patients with Cardiometabolic Disease" is an interesting study investigating the relationship between dietary fat intake and all cause mortality among patients with cardiometabolic syndrome, using data from NHANES cohorts. 

The reviewer would like to offer the following points for consideration:

1. Typically when we refer to the cardiometabolic pathology we refer to it as a syndrome as opposed to a disease, since there are no clear diagnostic criteria. The reviewer would suggest revising the term in the title and the text accordingly. If the authors would like to refer to it as a disease it is important to specify the diagnostic criteria used for such a qualifier.

2. Consider stating a clear hypothesis at the end of the introduction section.

3. The authors should consider delineating the inclusion criteria.

4. The first paragraph of the results section is essentially a rather long and difficult to follow sentence. Needs to be rewritten, in good English, broken down to shorter sentences and make references to the corresponding data (tables) for every result finding presented.

5. How was the follow-up time of 7.7 (as median) selected? What was the rationale?

6. BMI is an index and as such it doesn't have units. The division of weight in kg by height in meters squared is for calculation purposes only and does not refer to units since from a physical sense these units would refer to pressure (we are not measuring surface area obviously).

7. Please specify what were the cutoffs for alcohol consumption and smoking status.

8. What is meant by "insufficient" physical activity on table 1?

9. Was the intake of Fatty Acids normalized as per the total energy intake, fat energy intake and age? It would be interesting to do that and see if the analysis would yield different results.

10. How is the all cause mortality compare with quality of T2D management? T2D status and/or insulin resistance are essentially confounding factors. Did the authors consider these in their analyses?

11. Would the authors consider looking at food frequency questionnaires in order to verify trends and accuracy of dietary intake obtained information?

12. The conclusions section seems to be missing. The reviewer would suggest the inclusion of a conclusions section.  

13. Overall proofreading and run through of the manuscript is suggested. The reviewer would also suggest a native English speaker to run through the manuscript in order for the language to be improved for grammar, syntax and the length of the sentences to confer messages more clearly and with less confusion. Several sentences are fairly cumbersome to read.  

Round 2

Reviewer 1 Report

 I have no further comments.

Reviewer 3 Report

The authors have made a reasonable effort in addressing reviewer's comments. Proofreading is suggested.